# Comparison of Rapid Nucleic Acid Extraction Methods for SARS-CoV-2 Detection by RT-qPCR

**DOI:** 10.3390/diagnostics12030601

**Published:** 2022-02-27

**Authors:** Lívia Mara Silva, Lorena Rodrigues Riani, Marcelo Silva Silvério, Olavo dos Santos Pereira-Júnior, Frederico Pittella

**Affiliations:** Faculdade de Farmácia, Universidade Federal de Juiz de Fora, Juiz de Fora 36036-900, Brazil; liviamarasilva@gmail.com (L.M.S.); lorena.riani@ufjf.br (L.R.R.); marcelo.silverio@ufjf.edu.br (M.S.S.); olavo.pereira@farmacia.ufjf.br (O.d.S.P.-J.)

**Keywords:** SARS-CoV-2, COVID-19, RT-qPCR, rapid nucleic acid extraction

## Abstract

Since 2020, humanity has been facing the COVID-19 pandemic, a respiratory disease caused by the SARS-CoV-2. The world’s response to pandemic went through the development of diagnostics, vaccines and medicines. Regarding diagnostics, an enormous challenge was faced due to shortage of materials to collect and process the samples, and to perform reliable mass diagnosis by RT-qPCR. In particular, time-consuming and high cost of nucleic acid extraction procedures have hampered the diagnosis; moreover, several steps in the routine for the preparation of the material makes the extracted sample susceptible to contamination. Here two rapid nucleic acid extraction reagents were compared as extraction procedures for SARS-CoV-2 detection in clinical samples by singleplex and multiplex RT-qPCR analysis, using different transport media, samples with high and low viral load, and different PCR machines. As observed, rapid nucleic acid extraction procedures can be applied for reliable diagnosis using a TaqMan-based assay, over multiple platforms. Ultimately, prompt RNA extraction may reduce costs with reagents and plastics, the chances of contamination, and the overall time to diagnosis by RT-qPCR.

## 1. Introduction

COVID-19, caused by SARS-CoV-2, became a global pandemic disease from March of 2020, when the World Health Organization (WHO) declared a “public health emergency of international concern” [1]. The first case was reported in 2019 in the city of Wuhan, China, after the emergence of symptoms similar to pneumonia in part of the population [2]. The most common clinical symptoms of COVID-19 are fever, cough, fatigue, chest pain, headache and shortness of breath, characteristic of other viral illnesses such as the flu [3]. In December 2021, there were over 200 million confirmed cases and over 5 million deaths worldwide from COVID-19 [4].

The virus that causes the disease belongs to the Coronaviridae family, which has an RNA genome. Genetic sequencing showed that SARS-CoV-2 is 79% equivalent to SARS-CoV and 50% to MERS-CoV [1]. Thus far, there is no defined treatment to eliminate the virus, but vaccines are already available to control the disease progress [5,6,7]; however, the transmission of SARS-CoV-2 is very fast, and the rate of vaccination is usually slow in several countries, allowing new variants of the virus to emerge around the world [8]. The diagnosis of the disease allows both the correct management of the patient and the biosafety strategic actions to avoid the dissemination of the disease [9].

The most used methods for detecting or quantifying viruses nucleic acids in human biological samples are reverse transcription quantitative polymerase chain reaction (RT-qPCR) or real-time PCR. This technique detects a small segment of the viral genome, enabling rapid, sensitive and accurate strain-level detection of up to five targets in an assay. Several RT-qPCR assays have been designed for the detection of SARS-CoV-2 [10,11]. Currently, there are many WHO-approved commercial RT-qPCR kits for detecting SARS-CoV-2. In general, viral RNA is transcribed into cDNA, following the binding of the primers to newly synthesized cDNA. The primers usually target sequences that codify essential proteins from SARS-CoV-2, including the genes E, S, N, RdRp and Orf1ab [12]. Although it is a sensitive and accurate method for detecting viral genomes, it requires expensive sets of reagents, the availability of equipment and time to process the samples; moreover, there are several steps susceptible to mistakes that may impair the diagnosis of SARS-CoV-2, including sample collecting procedure and nucleic acid extraction [13].

One of the bottlenecks to the diagnosis of COVID-19 by RT-qPCR is the limited availability of RNA extraction kits for preparing the samples collected from patients. The extraction is usually time-consuming and becomes a problem due to the urgency required for diagnosis [14]; moreover, except for the major institutions, most of the diagnostic laboratories rely on manual extraction procedures which are prone to contamination, use large amounts of plastic materials (microtubes and pipette tips), or use toxic organic solvents. Thanks to the efforts of the scientific community, several nucleic acid extraction kits have been developed or improved. In this article, rapid nucleic acid extraction methods which speeds up SARS-CoV-2 detection by RT-qPCR were analyzed. In addition, the performance of two rapid nucleic acid extraction reagents, the Pi-Lise Nucleic Acid Extraction Reagent (Pi-Biotech Genética Avançada, Santos Dumont, MG, Brazil), and the QuickExtract DNA Extraction Solution (Lucigen Corporation, Middleton, WI, USA), were compared on biological samples for investigation of COVID-19.

## 2. Materials and Methods

### 2.1. Materials

In this study, two rapid nucleic acid extraction reagents were used: Pi-Lise Nucleic Acid Extraction Reagent, by Pi-Biotech Genética Avançada Ltda (Santos Dumont, MG, Brazil), and QuickExtract^TM^ DNA Extraction Solution, by Lucigen Corporation (Middleton, WI, USA). Once thawed, aliquots were stored at −20 °C to avoid >3 freeze–thaw cycles.

To perform the RT-qPCR assays, a multiplex kit (Allplex SARS-CoV-2 Assay, by Seegene Inc., Seoul, Republic of Korea), and a singleplex set of reagents [iTaq Universal Probes One-Step Kit, by Bio-Rad Laboratories (Hercules, CA, USA),were used, together with primers and controls 2019-nCoV RUO by IDT Integrated DNA Technologies (Coralville, IA, USA)]. The analyses were run in two distinct equipment: QuantStudio^™^ 3 and 7500Fast Real-Time PCR Systems, by Applied Biosystems (Foster, CA, USA).

### 2.2. Biological Samples

Biological samples for this study were collected from patients suspected of COVID-19 between August 2020 and August 2021. A total of 50 respiratory samples (nasopharyngeal swab or endotracheal aspirate in viral transport medium or saline solution) with known results emitted by local diagnostic laboratories were included in the study. The positive samples were chosen to present both high and low viral load (CT < 30.0 and CT > 30.0, respectively).

The study was submitted and approved by the local ethics committee (CEP UFJF—CAAE: 48021321.3.0000.5147) and included written consent from all participants.

### 2.3. Nucleic Acid Extraction

The method used to perform the extraction of nucleic acids from the samples was based in Ladha et al. [15], with slight modifications according to the supplier’s protocols. Important to note, viscous samples may require a pre-dilution with saline (NaCl 0.9%) prior to the extraction.

For the extraction, first the biological samples in viral transport medium (VTM) or saline (NaCl 0.9%) were diluted 1:1 with the extraction reagents. Here, a volume of 40 µL of sample was diluted with 40 µL of Pi-Lise or QuickExtract (Figure 1) and homogenized. Then, the mixture was heated to 95 °C for 5 min in a dry block and incubated at −20 in a freezer for 2 min. Samples were then thawed at room temperature, quick spinned (10 s) and subjected to RT-qPCR assays.

### 2.4. RT-qPCR Assay

The real-time PCR assay was performed using multiplexed and singleplexed approaches: the Allplex™ SARS-CoV-2 Assay (Seegene Inc., Seoul, Republic of Korea), and iTaq Universal Probes One-Step Kit (Bio-Rad Laboratories, Hercules, CA, USA), with primers, probes and controls 2019- nCoV RUO (IDT Integrated DNA Technologies).

Allplex™ SARS-CoV-2 Assay is a multiplexed RT-qPCR assay that detects and identifies in a single sample three target: RdRp and N genes (specific for SARS-CoV-2) and gene E (for all Sarbecoviruses, including SARS-CoV-2) [16,17]. On the other hand, the IDT panel corresponds to the primers and probes developed by the US CDC (Centers for Disease Control and Prevention, Atlanta, GA, USA); it was developed for singleplex analysis for RT-qPCR, with the detection of three genes: regions of the virus nucleocapsid gene (N1 and N2 targets) and an additional primer–probe set to detect the human RNase P gene (RP) as endogenous control [18]. It was combined with iTaq Universal Probes One-Step Kit (Bio-Rad Laboratories) for the singleplex analysis.

The real-time PCR was carried out in the 7500 Fast Real-Time (Applied Biosystems) for the multiplex assay. The final 25 μL reaction mixture contained 3.0 µL of extracted sample, 5.0 µL of primers and probes, 7.0 μL of enzyme mix and reaction buffer and 10.0 μL of nuclease-free water. The thermocycling conditions consisted of 50 °C for 20 min for reverse transcription, followed by 95 °C for 15 min and then 45 cycles of 95 °C for 15 s, 58 °C for 30 s.

The following reactions were performed using Bio-Rad Laboratories singleplex kit and IDT primers in a QuanStudio3 Real-Time PCR System (Applied Biosystems). The final 20 µL reaction mixture contained 3.0 µL of extracted sample, 0.5 μL of iScript enzyme mix, 10 μL of 2x iTaq PCR reaction mix and the combination of primers/probes and water (for N1 and RNaseP: 0.8 µL of primers and probes and 5.7 µL of nuclease-free water; for N2: 0.47 µL of primers and probes, and 6.03 of µL of nuclease-free water). Cycling was performed at 50 °C for 15 min for reverse transcription, followed by 95 °C for 3 min and then 40 cycles of 95 °C for 15 s, 60 °C for 1 min [19].

All runs included a positive SARS-CoV-2 genomic control and a negative control for the PCR amplification step. Fluorescence measurements were taken, and the threshold cycle (CT) value was calculated by determining the point at which fluorescence exceeded a defined threshold of the mean plus 10 standard deviations above baseline [20,21], according to the manufacturer’s recommendation. Clinical specimens were considered positive if two or more of the SARS-CoV-2 genomic targets showed positive results and all positive and negative control reactions resulted as expected.

### 2.5. Evaluation of the Influence of Pi-Lise Extraction Reagent on RT-qPCR

To investigate the interference of Pi-Lise reagent on RT-qPCR activity, RT-qPCR reactions using SARS-CoV-2 synthetic gene fragment dissolved in nuclease-free water or in a 50:50 nuclease-free water:Pi-Lise mixture were compared, as described by Ladha et al. for QuickExtract reagent [15]. Each RT-qPCR reaction was prepared using the Seegene kit in 7500 Fast system, and a total volume of 25 µL (3.0 µL of extracted sample, 5.0 µL of primers and probes, 7.0 μL of enzyme mix and reaction buffer and 10.0 μL of nuclease-free water).

### 2.6. Statistical Analysis

Statistical analysis was performed using GraphPad Prism version 8.0.1 (USA). Differences in the mean of cycle threshold (CT) values of genes were compared between the rapid extraction reagents by the T-Student test. Values with *p* < 0.05 were considered statistically significant. Graphs were prepared using GraphPad Prism version 8.0.1 (USA).

## 3. Results

### 3.1. Comparison of Rapid Extraction Reagents

Initially, the analyses were performed using the Allplex kit (Seegene), on the 7500 Fast Real-Time. The results presented in Table 1 show that the analyses with Pi-Lise and QuickExtract were the same regarding the final outcome (detected or not detected) for individual samples; moreover, both reagents presented 100% compatibility to the original results emitted after first analysis for diagnosis. The original Ct values used for diagnosis are shown in Appendix A, and were obtained using PureLink RNA Mini Kit (Invitrogen).

The CT values for gene N were similar in most samples, without statistical variations between the mean of the values of the two reagents (Figure 2). For the analysis of gene E, QuickExtract showed lower CT values with a significant difference (*p* < 0.05). As for the RdRp gene, there was no statistical significant difference between the mean CT values of Pi-Lise and QuickExtract. Ultimately, the obtained results showed that Pi-Lise and QuickExtract were effective in providing nucleic acid from samples in the different media tested. The detection of genes N, E and RdRp was viable in biological samples regardless of the transport media used (viral transport medium, saline and clinical samples of endotracheal aspirate) (Table 1).

A singleplex analysis (Bio-rad kit, with IDT primer-probes) was also performed after the extraction with both reagents. For this analysis, the QuantStudio 3 equipment was used. The results show that there was no significant difference in the final outcome when the samples were processed with Pi-Lise or QuickExtract (Table 2 and Figure 3). The process including both reagents provided compatible results using a different equipment and singleplex kit. As in the multiplex kit analysis, the singleplex results also show that Pi-Lise and QuickExtract are effective for extracting nucleic acids with different media.

### 3.2. Evaluation of the Influence of Pi-Lise Extraction Reagent on RT-qPCR

A series of reagent dilutions were evaluated to identify one that would achieve efficient lysis of the enveloped virus while preserving the activity of the RT-qPCR reaction. The results (Figure 4) show that Pi-Lise at a final concentration of 6.0% *v/v* did not interfere with the RT-qPCR reaction.

## 4. Discussion

This study compared two commercially available reagents on the extraction of nucleic acid from biological samples for COVID-19 diagnostic by RT-qPCR. The reagents Pi-Lise and QuickExtract promoted rapid extraction of nucleic acids, without the application of columns and organic solvents [22], for the use in TaqMan-based RT-qPCR.

The results obtained in this study show that SARS-CoV-2 was identified in different clinical samples after nucleic acid extraction by both reagents used separately. In addition, Pi-Lise and QuickExtract were effective in extracting viral RNA from endotracheal aspirate. These results demonstrate that the reagents are effective in complex matrices.

In nasopharyngeal swab samples, transported in viral transport medium and/or saline solution, Pi-Lise and QuickExtract were also efficient for nucleic acid extraction from samples with high viral load (CT ≤ 30.0) and for samples with low viral load (CT ≥ 30.0). In a study by Komiazyk et al., difficulty in extracting RNA from samples with low viral load was reported, in particular, due to operational difficulties in conventional extraction techniques (by column or organic solvents) [23]. Due to several steps in the extraction using column or solvent, and also the final eluting step in which the final concentration is dependent on the water volume added, the viral RNA concentration may be reduced, impairing the detection by RT-qPCR. The result obtained in this work demonstrates that rapid extraction reagents are effective in samples with low viral load, such as samples S7, S26 and S37. Patients with a low viral load are usually in the initial or final stage of the disease, which is a serious problem in case of a wrong diagnosis. Analyzing the compatibility of the original results emitted for diagnosis and the result obtained using either Pi-Lise or QuickExtract in the process, there was full compatibility among the reagents and the original diagnosis (detected and not detected). This indicates that both fast extracting reagents were effective in promoting the lysis and extraction of the SARS-CoV-2 nucleic acids, while preserving it from nucleases for a successful and reliable PCR reaction.

Based on the CT values obtained, both reagents showed similar results for gene N. Gene N is part of the viral nucleocapsid and has been reported as a long region of the viral gene sequence, highly conserved among Sarbecoviruses [24]. According to previous studies, results from RT-q-PCR using the N gene were more sensitive than those using other genes [12]. Therefore, the similarity in the values obtained demonstrates that both reagents do not interfere in the enzyme reaction with the gene N, validating the use of Pi-Lise and QuickExtract reagents for the detection kits comprising gene N, based on TaqMan reaction.

For the elaboration of the E gene primer–probe, a region of the viral structural protein envelope was used [25]. Although the mean CT values for gene E were different between the reagents, there was no change in the final qualitative result. The RdRp primer–probe is also able to detect the viral RNA-dependent RNA polymerase sequence [26]. The results obtained for this gene showed that there was no significant difference between the reagents in the mean CT values, indicating similar sensitivity.

In the analysis using a singleplex test, the extraction process employing Pi-Lise or QuickExtract was also satisfactory for the final detection of SARS-CoV-2 in different samples. In these reactions, the panel of primers used was that of the US CDC, which contained the genes N1 and N2, specific to SARS-CoV-2 and the RNAseP gene, as endogenous control [18]. According to the results, the viral genes that are pivotal for diagnosis were fully compatible between reagents. The virus was identified in 40% samples from varied biological matrices and was also 100% compatible with the original results emitted for diagnosis. The CT values observed for the viral genes were similar in the extractions performed with Pi-Lise and QuickExtract, proving the efficiency of lysis and extraction of viral nucleic acids and demonstrating the stability of the reagents in RT-qPCR reactions.

The Pi-Lise interference analysis in the RT-qPCR reaction showed that the reagent does not harm the PCR assay. The reaction volume was 25 µL, with 3 µL of sample. Among the samples containing Pi-Lise:Water mixture (50:50), the largest volume of Pi-Lise analyzed was 1.5 µL (6.0% *v/v*). In these samples, amplification was not observed in samples without positive control and the virus was detected at all concentrations tested. This result shows that Pi-Lise does not interfere in RT-qPCR reactions.

Nucleic acid extraction methods are of great importance for the diagnosis of several diseases, including COVID-19 [27]. In addition to easing the extraction process and requiring less instrumentation, the rapid extraction reagents proved to be effective in extracting and preserving the genetic material, enabling its safe use in a subsequent step of RT-qPCR based on TaqMan. It is important to note that the rapid method reduces the time of the extraction process to up to 7 min, which is of great importance in diagnosis using molecular biology tools.

Despite the efforts of the academic community being enormous, the control of COVID-19 is still a considerable challenge. New mutations are appearing periodically, with new cycles of dissemination, presenting significant difficulties in detecting and controlling the virus. Therefore, adequate diagnostic processes will still be essential for the correct management of infected people. The faster the diagnosis, the better the virus-fighting response; thus, nucleic acid extraction reagents are great allies in the control of COVID-19.

## 5. Conclusions

Many laboratories around the world are already using rapid nucleic acid extraction reagents to promptly obtain nucleic acids for RT-qPCR assay of SARS-CoV-2. Fast RNA extraction reduces costs with reagent and plastics, reduces the overall time to diagnosis by RT-qPCR and consequently reduces the enormous pressure that the technicians suffer during routine assays. Our study demonstrates that the Pi-Lise and QuickExtract reagents are effective in extracting and preserving viral and human nucleic acids to further detection of SARS-CoV-2 virus by TaqMan-based RT-qPCR in biological samples from diverse matrices (nasopharynx swabs, salive and endotracheal aspirates), using different equipment and diagnostic sets. Reliable results can be obtained using rapid nucleic acid extraction reagents and methods, with a positive impact on time and cost of analysis.

## Figures and Tables

**Figure 1 diagnostics-12-00601-f001:**
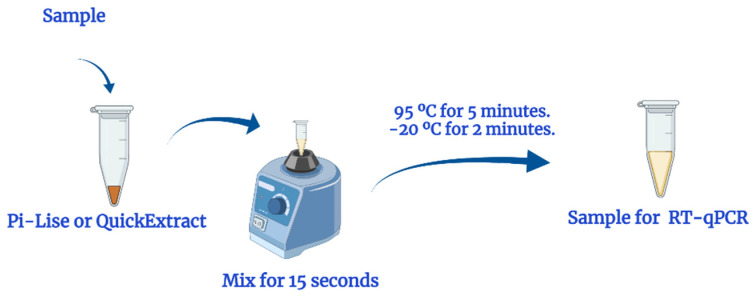
Scheme of rapid nucleic acid extraction.

**Figure 2 diagnostics-12-00601-f002:**
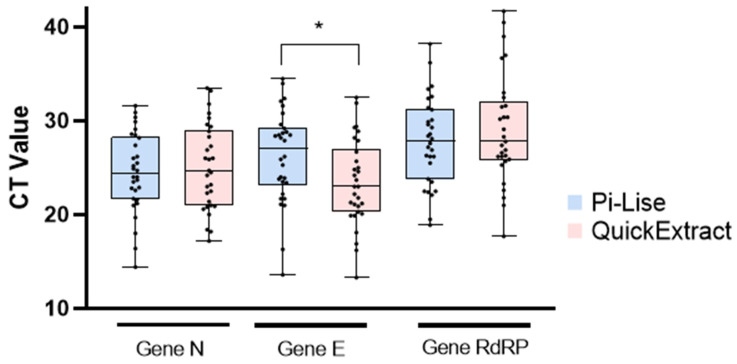
Cycle threshold (CT) values obtained for the genes N, E and RdRp in multiplex RT-qPCR assay after extraction with Pi-Lise and QuickExtract. The graph is presented as individual CT values, the mean and the dispersion (highest and lowest CT values). Statistical analysis was performed for the mean, where * indicates *p* < 0.05.

**Figure 3 diagnostics-12-00601-f003:**
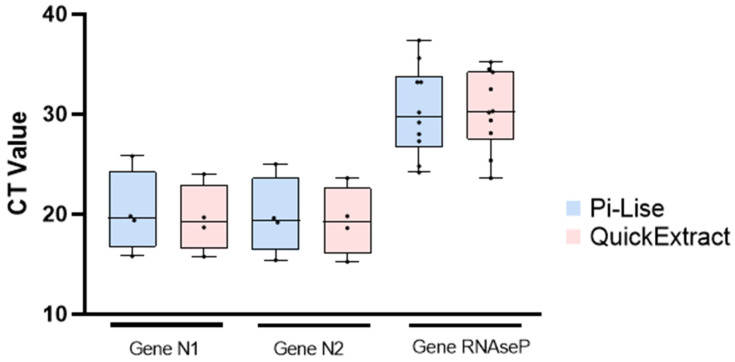
Cycle threshold (CT) values obtained for the genes N1, N2 and RNAseP in singleplex RT-qPCR assay after the extraction with Pi-Lise or QuickExtract. The graph is presented as individual CT values, the mean and the dispersion (highest and lowest Ct values). No statistically significant differences were observed between the reagents.

**Figure 4 diagnostics-12-00601-f004:**
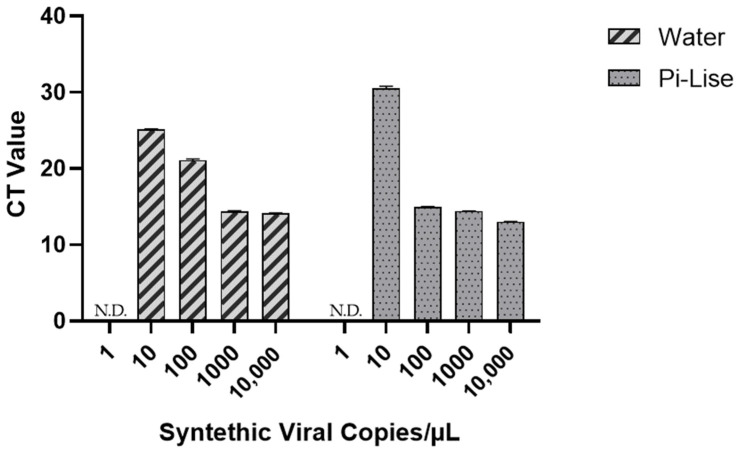
RT-qPCR of synthetic SARS-CoV-2 RNA control diluted in water or 50:50 H_2_O:Pi-Lise mixture. “N.D.” means not detected.

**Table 1 diagnostics-12-00601-t001:** RT-qPCR results of reactions performed with Allplex kit after extraction with different reagents (Pi-Lise or QuickExtract).

ID	Medium	Reagent	CT Value Gene N	CT Value Gene E	CT Value Gene RdRp	Result	Compatibility with Original Result for Diagnosis
S1	VTM	Pi-Lise	22.8	21.7	26.9	Detected	Yes
		QuickExtract	23.2	20.1	25.9	Detected	
S2	VTM	Pi-Lise	26.2	28.5	28.4	Detected	Yes
		QuickExtract	26.9	24.8	30.4	Detected	
S3	VTM	Pi-Lise	24.9	23.5	27.6	Detected	Yes
		QuickExtract	26.0	22.2	28.3	Detected	
S4	VTM	Pi-Lise	Undetermined	Undetermined	Undetermined	Not detected	Yes
		QuickExtract	Undetermined	Undetermined	Undetermined	Not detected	
S5	VTM	Pi-Lise	26.0	28.2	28.6	Detected	Yes
		QuickExtract	27.3	25.0	31.6	Detected	
S6	VTM	Pi-Lise	25.2	24.0	29.9	Detected	Yes
		QuickExtract	25.9	23.0	29.1	Detected	
S7	VTM	Pi-Lise	30.4	32.1	32.6	Detected	Yes
		QuickExtract	31.8	29.4	39.0	Detected	
S8	VTM	Pi-Lise	Undetermined	Undetermined	Undetermined	Not detected	Yes
		QuickExtract	Undetermined	Undetermined	Undetermined	Not detected	
S9	VTM	Pi-Lise	Undetermined	Undetermined	Undetermined	Not detected	Yes
		QuickExtract	Undetermined	Undetermined	Undetermined	Not detected	
S10	VTM	Pi-Lise	21.0	23.8	23.5	Detected	Yes
		QuickExtract	20.9	19.9	25.3	Detected	
S11	VTM	Pi-Lise	25.5	28.7	29.3	Detected	Yes
		QuickExtract	26.0	24.6	30.4	Detected	
S12	VTM	Pi-Lise	Undetermined	Undetermined	Undetermined	Not detected	Yes
		QuickExtract	Undetermined	Undetermined	Undetermined	Not detected	
S13	VTM	Pi-Lise	24.7	28.4	29.6	Detected	Yes
		QuickExtract	24.7	24.2	30.2	Detected	
S14	VTM	Pi-Lise	Undetermined	Undetermined	Undetermined	Not detected	Yes
		QuickExtract	Undetermined	Undetermined	Undetermined	Not detected	
S15	VTM	Pi-Lise	28.6	27.9	32.4	Detected	Yes
		QuickExtract	29.6	25.7	33.0	Detected	
S16	VTM	Pi-Lise	14.4	13.6	26.3	Detected	Yes
		QuickExtract	20.0	13.3	25.7	Detected	
S17	VTM	Pi-Lise	Undetermined	Undetermined	Undetermined	Not detected	Yes
		QuickExtract	Undetermined	Undetermined	Undetermined	Not detected	
S18	VTM	Pi-Lise	29.1	32.4	33.7	Detected	Yes
		QuickExtract	33.2	32.5	41.7	Detected	
S19	VTM	Pi-Lise	Undetermined	Undetermined	Undetermined	Not detected	Yes
		QuickExtract	Undetermined	Undetermined	Undetermined	Not detected	
S20	VTM	Pi-Lise	Undetermined	Undetermined	Undetermined	Not detected	Yes
		QuickExtract	Undetermined	Undetermined	Undetermined	Not detected	
S21	VTM	Pi-Lise	Undetermined	Undetermined	Undetermined	Not detected	Yes
		QuickExtract	Undetermined	Undetermined	Undetermined	Not detected	
S22	VTM	Pi-Lise	18.0	21.7	26.2	Detected	Yes
		QuickExtract	20.6	19.9	26.2	Detected	
S23	VTM	Pi-Lise	Undetermined	Undetermined	Undetermined	Not detected	Yes
		QuickExtract	Undetermined	Undetermined	Undetermined	Not detected	
S24	VTM	Pi-Lise	24.0	26.2	27.2	Detected	Yes
		QuickExtract	24.5	23.7	27.8	Detected	
S25	VTM	Pi-Lise	27.4	29.6	30.1	Detected	Yes
		QuickExtract	28.3	26.7	31.5	Detected	
S26	VTM	Pi-Lise	28.5	30.8	31.4	Detected	Yes
		QuickExtract	30.3	29.3	36.7	Detected	
S27	VTM	Pi-Lise	21.6	22.2	22.5	Detected	Yes
		QuickExtract	21.4	21.8	22.6	Detected	
S28	VTM	Pi-Lise	22.6	25.3	25.5	Detected	Yes
		QuickExtract	22.5	21.3	26.2	Detected	
S29	VTM	Pi-Lise	22.9	23.9	26.2	Detected	Yes
		QuickExtract	23.0	21.2	26.3	Detected	
S30	VTM	Pi-Lise	19.7	21.1	22.1	Detected	Yes
		QuickExtract	18.2	16.9	21.0	Detected	
S31	VTM	Pi-Lise	24.0	29.2	23.8	Detected	Yes
		QuickExtract	22.3	23.0	26.9	Detected	
S32	VTM	Pi-Lise	21.2	25.9	19.5	Detected	Yes
		QuickExtract	18.4	18.1	23.3	Detected	
S33	VTM	Pi-Lise	Undetermined	Undetermined	Undetermined	Not detected	Yes
		QuickExtract	Undetermined	Undetermined	Undetermined	Not detected	
S34	VTM	Pi-Lise	Undetermined	Undetermined	Undetermined	Not detected	Yes
		QuickExtract	Undetermined	Undetermined	Undetermined	Not detected	
S35	VTM	Pi-Lise	23.5	28.5	22.5	Detected	Yes
		QuickExtract	21.0	21.1	26.6	Detected	
S36	VTM	Pi-Lise	28.2	28.8	31.2	Detected	Yes
		QuickExtract	28.9	28.9	32.5	Detected	
S37	VTM	Pi-Lise	31.6	34.0	36.2	Detected	Yes
		QuickExtract	33.5	31.9	Undetermined	Detected	
S38	VTM	Pi-Lise	29.9	34.5	33.4	Detected	Yes
		QuickExtract	29.4	28.2	37.0	Detected	
S39	VTM	Pi-Lise	Undetermined	Undetermined	Undetermined	Not detected	Yes
		QuickExtract	41.5	Undetermined	Undetermined	Not detected	
S40	VTM	Pi-Lise	Undetermined	Undetermined	Undetermined	Not detected	Yes
		QuickExtract	Undetermined	Undetermined	Undetermined	Not detected	
S41	VTM	Pi-Lise	41.8	Undetermined	Undetermined	Not detected	Yes
		QuickExtract	Undetermined	Undetermined	Undetermined	Not detected	
S42	VTM	Pi-Lise	30.9	31.6	38.2	Detected	Yes
		QuickExtract	30.8	27.9	40.5	Detected	
S43	VTM	Pi-Lise	Undetermined	Undetermined	Undetermined	Not detected	Yes
		QuickExtract	Undetermined	Undetermined	Undetermined	Not detected	
S44	VTM	Pi-Lise	23.7	23.4	28.1	Detected	Yes
		QuickExtract	24.2	20.9	27.4	Detected	
S45	Saline	Pi-Lise	Undetermined	Undetermined	Undetermined	Not detected	Yes
		QuickExtract	Undetermined	Undetermined	Undetermined	Not detected	
S46	Saline	Pi-Lise	Undetermined	Undetermined	Undetermined	Not detected	Yes
		QuickExtract	Undetermined	Undetermined	Undetermined	Not detected	
S47	Saline	Pi-Lise	16.4	16.3	18.9	Detected	Yes
		QuickExtract	17.2	16.2	17.7	Detected	
S48	Endotracheal aspirate	Pi-Lise	21.7	21.0	22.4	Detected	Yes
		QuickExtract	20.8	20.3	21.8	Detected	
S49	Endotracheal aspirate	Pi-Lise	Undetermined	Undetermined	Undetermined	Not detected	Yes
		QuickExtract	Undetermined	Undetermined	Undetermined	Not detected	
S50	Endotracheal aspirate	Pi-Lise	Undetermined	Undetermined	Undetermined	Not detected	Yes
		QuickExtract	Undetermined	Undetermined	Undetermined	Not detected	

CT rt-qPCR = Cycle threshold.

**Table 2 diagnostics-12-00601-t002:** RT-qPCR results of reactions performed with Bio-rad kit and IDT primers-probes after extraction with different reagents (Pi-Lise or QuickExtract).

ID	Medium	Reagent	CT Value Gene N1	CT Value Gene N2	CT Value Gene RNAseP	Result	Compatibility with Original Result for Diagnosis
S51	VTM	Pi-Lise	19.4	19.2	35.6	Detected	Yes
		QuickExtract	19.7	19.8	25.4	Detected	
S52	VTM	Pi-Lise	25.8	25.0	37.4	Detected	Yes
		QuickExtract	24.0	23.6	28.1	Detected	
S53	VTM	Pi-Lise	Undetermined	Undetermined	27.3	Not Detected	Yes
		QuickExtract	Undetermined	Undetermined	34.5	Not Detected	
S54	VTM	Pi-Lise	Undetermined	Undetermined	28.0	Not Detected	Yes
		QuickExtract	Undetermined	Undetermined	34.2	Not Detected	
S55	Saline	Pi-Lise	Undetermined	Undetermined	30.2	Not Detected	Yes
		QuickExtract	Undetermined	Undetermined	35.2	Not Detected	
S56	Saline	Pi-Lise	Undetermined	Undetermined	29.2	Not Detected	Yes
		QuickExtract	Undetermined	Undetermined	30.3	Not Detected	
S57	Saline	Pi-Lise	15.8	15.4	33.2	Detected	Yes
		QuickExtract	15.8	15.2	29.4	Detected	
S58	Endotracheal aspirate	Pi-Lise	19.8	19.6	33.2	Detected	Yes
		QuickExtract	18.7	18.6	23.6	Detected	
S59	Endotracheal aspirate	Pi-Lise	Undetermined	Undetermined	24.8	Not Detected	Yes
		QuickExtract	Undetermined	Undetermined	32.5	Not Detected	
S60	Endotracheal aspirate	Pi-Lise	Undetermined	Undetermined	24.2	Not Detected	Yes
		QuickExtract	Undetermined	Undetermined	30.2	Not Detected	

CT rt-qPCR = Cycle threshold.

## Data Availability

All relevant data are within the manuscript.

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
