# Peer review of "Comparison of Rapid Nucleic Acid Extraction Methods for SARS-CoV-2 Detection by RT-qPCR"

_diagnostics, 2022, doi:10.3390/diagnostics12030601_

Round 1

Reviewer 1 Report

The manuscript describes a comparison of two rapid nucleic acid extraction methods applied in two different RT-qPCR assays, using different transport media, high and low viral load samples and different PCR machines.

The Authors reported CT values of samples in which SARS-CoV-2 genetic material was detected. The study showed that qualitatively there was no difference between the nucleic acid extraction methods used and the kind of tests. Is there a quantitative difference?

Furthermore, please demonstrate with statistical methods that there is no significant statistical difference between the RNA isolation methods or tests.

I also have reservations about the language, usually in scientific articles we do not write in the personal form. Authors often use the "we" form (e.g. points 2.5; 3.2). Please rewrite the entire text and change the sentences to avoid the personal form.

Reviewer 2 Report

The manuscript presented by Silva and co-authors is a well written comparison between two rapid nucleic acid extraction methods for SARS-CoV-2 detection with RT-qPCR. Although there is not much novelty in this report, since commercially available kit for rapid nucleic acid extraction and SARS-CoV-2 detection are on the market since the beginning of 2021, it is indisputable the importance of such systems for decreasing the pressure on analytical laboratories and the whole healthcare system. 

The data is well presented and the methods used are clearly described, however the reviewer has a concern regarding the study design. In particular:

1) It appears that there is consistency between the two fast extraction methods and the original diagnostic results. However, the reviewer has the opinion that  it would have been interesting to compare the Ct values differences (if any) between the analysis with the two rapid extraction kits and the regular viral isolation system. This would have better clarified if a loss of sensitivity could be expected. Could the authors explain why in their study design was not included a comparison with a regular extraction system?

2) The sample with the lowest viral load identified had a Ct value of 31,6. Given the missing info of the originally identified Ct values, it is hard to understand whether higher Ct values would have been detected or not with these rapid systems. This would be important to know, since commercially available kits for SARS-CoV-2 RT-qPCR detection are able to identify virus presence up to Ct values of 37, and in most cases is important for early identification of infection. Could the author give the information for the viral load obtained in the original result for diagnosis? 

3) Following the question at point 2, did the authors analysed positive diagnosed samples with Ct values higher than 32? This would help elucidate whether these two rapid extraction systems would maintain the same sensitivity as the regular extraction system used for diagnosis.

4) A clear definition of the threshold used for establishing if a genomic target was positive is missing. Could the author add such information in the method section?

Round 2

Reviewer 2 Report

The reviewer has no further questions following the amendments done by the authors.